# Patient costs for prevention of mother-to-child HIV transmission and antiretroviral therapy services in public health facilities in Zimbabwe

Innocent Chingombe[1]*, Munyaradzi P. Mapingure[1], Shirish Balachandra[2], Tendayi N. Chipango[1], Fiona Gambanga[1], Angela Mushavi[3], Tsitsi Apollo[3], Chutima Suraratdecha[4], John H. Rogers[2], Leala Ruangtragool[5], Elizabeth Gonese[2], Godfrey N. Musuka[1], Owen M. Mugurungi[3], Tiffany G. Harris[1]

1 ICAP at Columbia University, New York, NY, United States of America, 2 U.S. Centers for Disease Control and Prevention, Harare, Zimbabwe, 3 Ministry of Health and Child Care, Harare, Zimbabwe, 4 U.S. Centers for Disease Control and Prevention, Atlanta, Georgia United States of America, 5 PHI/CDC Global HIV Surveillance Fellow, U.S. Centers for Disease Control and Prevention, Harare, Zimbabwe

* ic2421@cumc.columbia.edu

**Data Availability Statement:** All relevant data are within the paper and its Supporting Information

## Abstract

Zimbabwe has made large strides in addressing HIV. To ensure a continued robust response, a clear understanding of costs associated with its HIV program is critical. We conducted a cross-sectional evaluation in 2017 to estimate the annual average patient cost for accessing Prevention of Mother-To-Child Transmission (PMTCT) services (through antenatal care) and Antiretroviral Treatment (ART) services in Zimbabwe. Twenty sites representing different types of public health facilities in Zimbabwe were included. Data on patient costs were collected through in-person interviews with 414 ART and 424 PMTCT adult patients and through telephone interviews with 38 ART and 47 PMTCT adult patients who had missed their last appointment. The mean and median annual patient costs were examined overall and by service type for all participants and for those who paid any cost. Potential patient costs related to time lost were calculated by multiplying the total time to access services (travel time, waiting time, and clinic visit duration) by potential earnings (US$75 per month assuming 8 hours per day and 5 days per week). Mean annual patient costs for accessing services for the participants was US$20.00 [standard deviation (SD) = US$80.42, median = US$6.00, range = US$0.00–US$12,18.00] for PMTCT and US$18.73 (SD = US $58.54, median = US$8.00, range = US$0.00–US$ 908.00) for ART patients. The mean annual direct medical costs for PMTCT and ART were US$9.78 (SD = US$78.58, median = US$0.00, range = US$0.00–US$ 90) and US$7.49 (SD = US$60.00, median = US$0.00) while mean annual direct non-medical cost for US$10.23 (SD = US$17.35, median = US $4.00) and US$11.23 (SD = US$25.22, median = US$6.00, range = US$0.00–US$ 360.00). The PMTCT and ART costs per visit based on time lost were US$3.53 (US$1.13 to US $8.69) and US$3.43 (US$1.14 to US$8.53), respectively. The mean annual patient costs per person for PMTCT and ART in this evaluation will impact household income since PMTCT and ART services in Zimbabwe are supposed to be free.

files. Data can also be accessed from third parties in writing, from the MOHCC Epidemiologist Dr Brian Moyo, who is based at MOHCC AIDS and TB Programme office at 2nd floor Mkwati Building in Harare, via the following e-mail, moyobk1@gmail.com. Phone +263 774021723.

**Funding:** This project has been supported by the President's Emergency Plan for AIDS Relief (PEPFAR) through the Centers for Disease Control and Prevention (CDC) under the terms of cooperative agreement "Strengthening Epidemiology and Strategic Information in the Republic of Zimbabwe under PEPFAR" (U2GGH001939-01).

**Competing interests:** The authors have declared that no competing interests exist.

## Introduction

Access to prevention of mother-to-child transmission of HIV (PMTCT) and antiretroviral therapy (ART) services has been greatly expanded in Zimbabwe. In 2015, there were an estimated 1.4 million people living with HIV (PLHIV) in Zimbabwe, including 68,000 pregnant women [1]. By the end of December 2015, a total of 1,560 sites were offering ART services to 879,271 individuals, translating to about 63% of the total estimated number of PLHIV in the country [2].

The rise in treatment costs for individuals and families with chronic health conditions can impose a significant financial burden among even those with health insurance [3]. One review found that catastrophic health expenditures and impoverishment were highest for persons living with HIV receiving ART in Sub-Saharan African countries [4]. In Zimbabwe, even though HIV treatment is provided free of charge at public facilities, treatment of opportunistic infections and access to other diagnostic services may not be covered or available, forcing patients to seek care at private facilities resulting in expenses for patients.

To ensure a continued robust HIV response in Zimbabwe and to achieve HIV elimination, a clear understanding of program and patient costs associated with HIV are critical. Patient costs also include transport costs incurred when going to access services and earnings the patient did not receive while visiting the health facility. Against this background we evaluated both service provider and patient costs associated with receiving HIV services in Zimbabwe. In this paper we focus on the patient costs.

## Methods

### Ethical approval

The evaluation protocol received human subjects approval from the Medical Research Council of Zimbabwe and the Columbia University Institutional Review Board. The protocol was also reviewed in accordance with the United States Centers for Disease Control and Prevention (CDC) and determined to be a non-research, program evaluation.

### Facilities

This evaluation included health facilities with both PMTCT services through antenatal care (ANC) and an ART clinic. In Zimbabwe these are different points of services and pregnant women with HIV receive ART in ANC until they are 24 months postpartum. Other adults with HIV receive HIV-related services through the ART clinic. Of the 1,560 PMTCT/ART co-located health facilities in Zimbabwe at the time of the evaluation, we eliminated 565 those that were for-profit, served closed populations such as uniformed forces and miners, or had less than 12 months of experience providing comprehensive PMTCT/ART services at the start of data collection. Additionally, facilities were excluded if they were experiencing difficulties with program implementation as determined by members of the project team or if they had low patient volumes (<20 ART patients), as these facilities were generally not representative. Of the 995 remaining facilities, 20 were selected using stratified random sampling. Eligible facilities were first categorized by service delivery level (primary, secondary, tertiary, or quaternary) then categorized by geographic area (northern and southern regions). One health facility was then randomly selected from each facility level within each geographic area.

### Participants

Data were collected from September to November 2017. At each site, we selected a cross-sectional convenience sample among adult patients (aged ≥18 years) receiving either outpatient

PMTCT services in ANC (pregnant women and postpartum women with HIV) or ART services (ART and pre-ART patients) at the facility on the day of data collection and for whom the date of encounter was not the first visit at the facility. Individuals were not eligible for participation if they were aged less than 18 years of age; not enrolled in the PMTCT or ART programs at the site; newly registered PMTCT or ART patients (to ensure that individuals in the sample have had experience with PMTCT or ART services in the facility); did not speak English, Shona, or Ndebele; or did not provide consent. Because HIV-related services are different for pregnant and postpartum women than for other PLHIV likely resulting in different costs to both the clients and the program and their age and sex structure differ, we evaluated them separately from other adults receiving HIV services. Clinical staff referred potential participants to project staff for eligibility screening. Because participants enrolled on-site may visit the facility more frequently, resulting in overestimation of patient costs (since more frequent facility visits entail higher costs), the evaluation also included phone interviews for adult ART or PMTCT patients who had at least one prior appointment at the facility and had missed their most recent appointment. Health facility staff used clinic records to identify potential participants for the phone interviews and obtained verbal consent to provide contact information to project staff.

After eligibility was confirmed, participants were enrolled in person or via telephone and provided informed oral consent. For informed oral consent, interviewers read the consent form to eligible, potential participants in the language of their preference. Potential participants were provided the opportunity to ask any questions. The consent form covered all procedures, potential risks and benefits, and who to contact to report complaints or concerns. After potential participants clearly indicated that they understood the content of the informed consent form, they provided verbal consent by repeating the consenting statement. Those not willing to take part, would read a refusal statement. The interviewer signed an attestation that consent was provided. Those that were enrolled in person were provided a copy of the consent form. Patients were enrolled sequentially until sample size targets were met for each facility was reached. Across all facilities, the target sample sizes were 547 (420 on-site and 127 phone) PMTCT participants and 767 ART (460 on-site and 87 phone) participants. Patient-level data were collected during in-person and telephone interviews using structured questionnaires on tablets. Information collected included demographics and the amount of money (in US dollars, the official currency of Zimbabwe at the time of the evaluation) and time spent (e.g., for transportation, registration, medications, and laboratory testing) in the last 6 months to receive ART or PMTCT services.

## Cost estimates

We estimated the costs using micro-costing from patient level data using the patient level questionnaire where they reported direct medical, non-medical and indirect costs incurred from health-related visits over a six-month period prior to enrolment in the study. Direct medical costs referred to costs incurred for consultation, medication, diagnostic and monitoring tests while non-medical costs were related to travel, accommodation and other costs associated with the health-related visits. Indirect costs included opportunity costs associated with travel time and time spent in the health facility. The source of income was provided by participants in the questionnaire while the hourly rate was estimated by looking at the prevailing wage of that occupation as provided in the literature (see Table 1). We calculated the opportunity cost in three ways: 1) we used actual income estimates for those participants who received a wage from actual work and not passive or no income at all; 2) We allocated 1 to all the participants who had a passive income or not receiving an income at all and 3) we varied the income to the

**Table 1. Source and type of income.**

| Type of income | Amount | Source |
|---|---|---|
| Wage work | $0,56 | *Crush J & Chikanda A. (2016). Zimbabwe's Exodus: Crisis, migration, survival. Chapter 13:, Migrant Remittances and Household Survival in Zimbabwe. Accessed from: http://samponline.org/wp-content/uploads/2016/10/Zimbabwes-Exodus-Chapter-13.pdf* |
| Casual Work | $0,53 | |
| Remittance-money | $0,56 | |
| Remittance-goods | $0,53 | |
| Sale of farm products | $0,53 | |
| Formal business | $0,52 | |
| Informal business | $ 0,54 | |
| Pension | $0,53; $ 0,33; $0,23 | *Crush J & Chikanda A. (2016). Zimbabwe's Exodus: Crisis, migration, survival. Chapter 13:, Migrant Remittances and Household Survival in Zimbabwe. Accessed from: http://samponline.org/wp-content/uploads/2016/10/Zimbabwes-Exodus-Chapter-13.pdf*<br> *International Labour Office. (2012). Zimbabwe: Report to the Government: Actuarial Study on the National Pension Scheme / International Labour Office, Social Security Department.—Geneva: ILO, 2012 xiii. 51 p. accessed from: https://www.social-protection.org/gimi/RessourcePDF.action;jsessionid=LgRIIVvrJnZVnwRkjlz-4lLmwYfUuprHig5OiBn-J5OKslpvjBAv!474614842?id=31528*<br> *Social Security Administration. (2017). Social Security Programs Throughout the World: Africa, 2017, Zimbabwe. Accessed from: https://www.ssa.gov/policy/docs/progdesc/ssptw/2016-2017/africa/zimbabwe.html* |
| Gifts | $0,52 | *Crush J & Chikanda A. (2016). Zimbabwe's Exodus: Crisis, migration, survival. Chapter 13:, Migrant Remittances and Household Survival in Zimbabwe. Accessed from: http://samponline.org/wp-content/uploads/2016/10/Zimbabwes-Exodus-Chapter-13.pdf* |
| Private Wage | $2,95 | *Labour and Economic Development Research Institute of Zimbabwe (LEDRIZ). (2016) USAID Strategic Economic Research And Analysis–Zimbabwe (SERA) Program Wage Structure And Labour Costs In Zimbabwe: An Analysis Of Flexibility, Competitiveness And Equity. Accessed from https://www.ilo.org/wcmsp5/groups/public/—africa/—ro-abidjan/—sro-harare/documents/genericdocument/wcms_470742.pdf* |
| Public Wage | $3,73 | |
| Municipal Wage | $4,54 | |
| Parastatal Wage | $4,43 | |
| NGOs Wage | $5,19 | |
| Daily general worker (unskilled) | $0,64 | *DCLA. (2016). 3.3 Zimbabwe Manual Labor Costs, Accessed from https://dlca.logcluster.org/display/public/DLCA/3.3+Zimbabwe+Manual+Labor+Costs 2015* |
| Daily general worker (skilled) | $1,58 | |
| Skilled labour | $3,14 | |
| Catering | $1,15 | |
| Mining | $1,41 | |
| Harare municipal undertaking | $1,17 | |
| Property Income | $69,00 (monthly wage) | *ZIMSTAT. (2013) Poverty Income Consumption and Expenditure Survey 2011/12 Report. Accessed from https://www.zw.undp.org/content/zimbabwe/en/home/library/poverty/poverty-income-consumption-and-expenditure-and-survey-2011-12.html* |
| Minimal sectoral ncome | $0,42; $0,41 | *WageIndicator 2021 - ALREI.org - Minimum Wages And Collective Bargaining In Zimbabwe Compiled By Naome Chakanya, Ledriz, September 2016 Accessed from https://alrei.org/education/minimum-wages-and-collective-bargaining-in-zimbabwe-compiled-by-naome-chakanya-ledriz-september-*<br> *Zhangazha Wongai. Farm Workers are Suffering! [Internet]. Mywage.org/Zimbabwe, Wage Indicator Foundation . 2018 [cited 2018 Oct 5]. Available from: https://mywage.org/zimbabwe/main/salary/minimum-wage-1/farm-workers-are-suffering* |

sectoral minimal rate and the maximum casual wage rate. This income was given at 2016 rates. The currency used was 2016 United States Dollars at the December 2016 average exchange rate of 1 US Dollar: 13.38 South African Rands (ZAR) [5] where applicable. No discount rate was applied. Costs were stratified by the type of service the clients were receiving i.e. ART or PMTCT.

## Statistical analysis

Analysis was conducted using Stata MP 15 (StataCorp, College Station, TX) [6]. Mean annual patient costs were calculated, which included direct payments by patients for HIV-related

consumables and services and transportation costs related to obtaining those consumables and services. We also separately calculated patient costs related to income that individuals could have received if they had not visited the clinic.

The mean and median patient annual cost was calculated separately for PMTCT and ART for all participants regardless of services received and whether they paid for any services. The same calculations were repeated for participants who paid some cost. The patient costs for specific services associated with PMTCT and ART were also calculated among those who accessed that specific service or consumable. Standard deviations (SD) were calculated. For the individual services costs, we multiplied the amount of money spent by the patient by the number of times the patient accessed the service or consumable in the six months before the interview; this cost was doubled to obtain annual costs. At the time of the evaluation, most patients had an appointment every three months and differentiated service delivery models such as multi-month scripting had not been widely implemented.

Potential patient costs related to time lost were calculated by measuring the total time spent (including travel time, waiting time, and time receiving services) to access services reported by patients and multiplying that by the amount of money the patient could have earned. To determine the income we initially calculated hourly wages from available sources based on their primary sources of income, given that the income was not passive and then assumed a minimum monthly sectoral wage of US$75.00 for all patients [7] and a maximum casual rate of skilled workers of US$3.14 per hour [8]. For the latter calculations, we converted all the passive income and non-working responses to $1 to avoid dividing by 0 [9]. We assumed 22 days per month and an 8-hour workday. Zimbabwe does not have a minimum wage so US$75.00 was used as that was the wage that was proposed as the Zimbabwe 2018 farm workers minimum wage and discounted to 2016 dollars [10].

## Results

We interviewed 923 patients: 424 onsite PMTCT participants, 47 offsite PMTCT participants, 414 onsite ART participants, and 38 offsite ART participants. Due to active programs to return patients to care, sample size goals were met for onsite participants, but not for offsite participants, so all analyses combined the two groups.

Almost all PMTCT participants [467 of 471 (99%)] and ART participants [441 of 452 (98%)] reported that they were receiving ART. Most ART participants were women (63%). Mean ages of the respondents for PMTCT and ART participants were 30 years (SD, 7; range, 18–45 years) and 43 years (SD, 12; range, 18–73 years), respectively. Most participants in both groups had either some level of secondary education or had completed secondary education (Table 2).

Among all participants (regardless if they paid any costs), mean annual patient costs were US$20.00 (SD = US$80.42) per PMTCT patient (median = US$6.00, range = US$0.00–US$1218.00) and US$19.31 (SD, US$65.18) per ART patient (median = US$8.00, range = US$0.00–US$908.00). Among those who paid any cost (651 of 923 or 71%) mean annual patient costs were US$28.90 (SD = US$95.37) per PMTCT patient (median = US$12.00, range = US$1.00–US$1218.00) and US$26.86 (SD = US$75.56) per ART patient (median = US$12.00, range = US$1.00–US$908.00).

We also estimated participant annual costs incurred as part of accessing specific PMTCT services (Table 3) or ART services (Table 4) at any health facility and these included transport, antiretroviral drugs (ARV), initial appointments, routine appointment, CD4 tests, in-patient stays, and drugs other than ARV among participants who accessed that specific service or consumable. The highest annual cost among PMTCT participants was for in-patient stays (US

**Table 2. Self-reported demographic characteristics for participants at the 20 participating facilities, Zimbabwe, 2017.**

| Variable | PMTCT N (%) | ART N (%) |
|---|---:|---:|
| Number of patients | 471 | 452 |
| Location of interview | | |
| On-site | 424 (90%) | 414 (92%) |
| Off-site | 47 (10%) | 38 (8%) |
| Receiving ART | | |
| Yes | 467 (99%) | 441 (98%) |
| No | 4 (1%) | 11 (2%) |
| Gender | | |
| Male | N/A | 166 (37%) |
| Female | 471 (100%) | 286 (63%) |
| PMTCT status | | |
| Pregnant | 102 (22%) | 3* |
| Breastfeeding | 354 (78%) | 3** |
| Mean age (years) (SD) | 30 (7) | 43 (12) |
| Age range (years) | 18–45 | 18–73 |
| Education | | |
| No education | 3 (1%) | 13 (35%) |
| Some primary | 39 (8%) | 47 (10%) |
| Completed primary | 83 (18%) | 80 (18%) |
| Some secondary | 178 (38%) | 122 (27%) |
| Completed secondary | 140 (30%) | 142 (31%) |
| Any tertiary | 28 (6%) | 44 (10%) |
| Refused to answer | 0 (0%) | 4 (1) |
| Employment status | | |
| Farming or livestock keeping or fishing | 61 (13%) | 71 (16%) |
| Paid employee (Government or Private) | 69 (15%) | 121 (27%) |
| Self-employed | 88 (19%) | 106 (23%) |
| Not working | 66 (14%) | 61 (13%) |
| Home maker/housewife/house chores | 165 (35%) | 47 (10%) |
| Student | 3 (1%) | 7 (2%) |
| Other | 19 (4%) | 35 (8%) |
| Refused to answer | 0 (0%) | 4 (1%) |

*In the ART sample, there were 3 patients who reported being pregnant and were awaiting transfer to PMTCT and one patient who did not know her pregnancy status.

**3 patients in ART were still breastfeeding due to prolonged breastfeeding.

N/A: Not Applicable

SD: Standard Deviation

$216.36, n = 11), followed by other HIV treatment costs not listed above, such as full blood count and erythrocyte sedimentation rate testing (US$42.00, n = 1) and other government facility costs (US$27.66, n = 40). Mean annual costs for transportation to health facilities for PMTCT participants was US$10.49, n = 459. All other mean annual costs were <US$10.00, with the lowest being ARV at US$0.03, n = 465 per year. None of the PTMTCT participants reported any expenses related to viral load testing in the previous 6 months. For ART participants, the highest costs were inpatient admission (US$67.71, n = 7) and other private clinic costs incurred during ART treatment (US$58.59, n = 17) (Table 4). For both groups,

**Table 3. Estimated participant annual costs for receiving PMTCT services among participants who incurred costs associated with that service or consumable, Zimbabwe 2017[*].**

| Item/Service | Number of participants reporting receiving the service | Mean (US$) | Min (US$) | Max (US$) |
|---|---|---|---|---|
| Transport | 459 | $10.49 | $0.00 | $160.00 |
| ARVs | 465 | $0.03 | $0.00 | $12.00 |
| Initial registration appointment | 23 | $6.09 | $0.00 | $24.00 |
| Routine appointment | 60 | $7.93 | $2.00 | $32.00 |
| Emergency appointment | 5 | $7.20 | $2.00 | $18.00 |
| CD4 tests | 4 | $4.00 | $4.00 | $4.00 |
| Viral load tests | 0 | | | |
| X-rays | 3 | $6.67 | $0.00 | $20.00 |
| Inpatient stays | 11 | $216.36 | $24.00 | $1200.00 |
| Other costs | 1 | $42.00 | $42.00 | $42.00 |
| Other HIV-related drugs or supplies | 27 | $8.63 | $0.00 | $30.00 |
| Other government facility | 40 | $27.66 | $0.00 | $1040.00 |
| Other private clinic | 10 | $8.60 | $0.00 | $80.00 |
| Traditional healer | 9 | $0.00 | $0.00 | $0.00 |
| Pharmacy | 11 | $5.32 | $0.00 | $30.00 |
| Costs at other facilities | 8 | $0.00 | $0.00 | $0.00 |

[*]Reported costs for each service were multiplied by the number of times a patient utilized the service over 6 months and then that was doubled to reflect annual costs.

participants did not report any costs for traditional healer services or any costs for services at other health facilities (Tables 3 and 4).

The mean round-trip travel time per visit, including time waiting for transport, reported by PMTCT participants was 1 hour and 5 minutes (range, 2 minutes–8 hours 15 minutes). Mean waiting times at the facility to see a clinician were 44 minutes (range, 0–7 hours) for PMTCT

**Table 4. Estimated participant annual costs for receiving ART services among participants who incurred cost associated with that service or consumable, Zimbabwe 2017[*].**

| Service | Number of participants reporting receiving the service | Mean (US$) | Min | Max |
|---|---|---|---|---|
| Transport | 438 | $11.62 | $0.00 | $360.00 |
| ARVs | 439 | $0.17 | $0.00 | $72.00 |
| Initial registration appointment | 42 | $5.48 | $0.00 | $20.00 |
| Routine appointment | 44 | $6.70 | $0.80 | $12.00 |
| Emergency appointment | 7 | $7.00 | $1.00 | $18.00 |
| CD4 tests | 6 | $57.33 | $4.00 | $288.00 |
| Viral load tests | 1 | $24.00 | $24.00 | $24.00 |
| X-rays | 13 | $38.15 | $0.00 | $100.00 |
| Inpatient stays | 7 | $67.71 | $0.00 | $200.00 |
| Other costs | 4 | $13.50 | $4.00 | $26.00 |
| Other HIV-related drugs or supplies | 17 | $8.29 | $2.00 | $24.00 |
| Other government facility | 37 | $3.78 | $0.00 | $70.00 |
| Other private clinic | 17 | $58.59 | $0.00 | $900.00 |
| Traditional healer | 14 | $0.00 | $0.00 | $0.00 |
| Pharmacy | 14 | $5.00 | $0.00 | $40.00 |
| Costs at other facilities | 14 | $0.00 | $0.00 | $0.00 |

[*]Reported costs for each service were multiplied by the number of times a patient utilized the service over 6 months and then that was doubled to reflect annual costs.

**Table 5. Estimated average monthly income and disaggregation of costs by type of income.**

| Program | Monthly income | | | Direct Medical Costs | | | Direct non-medical costs | | | Total Medical costs | | | Indirect Costs (Opportunity costs using estimated income) | | |
|---------|------|--------|-----------------------|------|--------|-----------------------|------|--------|-----------------------|------|--------|-----------------------|------|--------|-----------------------|
| | Mean | Median | Standard deviation | Mean | Median | Standard deviation | Mean | Median | Standard deviation | Mean | Median | Standard deviation | Mean | Median | Standard deviation |
| PMTCT | $201.02 | $275.00 | $111.66 | $9.78 | $0.00 | $78.58 | $10.23 | $1400 | $17.35 | $20.00 | $6.00 | $80.42 | $3.53 | $4.14 | $25.83 |
| ART | $193.06 | $275.00 | $119.82 | $7.49 | $0.00 | $60.00 | $11.23 | $6.00 | $25.22 | $18.73 | $8.00 | $58.54 | $3.43 | $4.20 | $17.50 |
| Total | $197.13 | $275.00 | $115.73 | $8.66 | $0.00 | $66.49 | $10.71 | $6.00 | $21.56 | $19.38 | $8.00 | $70.52 | $3.48 | $3.48 | $4.17 |

participants with an appointment and 1 hour 1 minute (range, 0–5 hours) for those without an appointment. Average time spent seeing a clinician was 13 minutes (range, 0–2 hours). The mean time (including traveling and waiting time at the facility) for a PMTCT visit was 3 hours 11 minutes (range, 0–18 hours 45 minutes). For ART participants, the mean travel time was 1 hour 4 minutes (range, 0–10 hours). Mean waiting times to see a clinician were 47 minutes (range, 0–5 hours 15 minutes) for ART participants with an appointment and 53 minutes (range, 0–5 hours) for those without an appointment. The mean time spent seeing a clinician was 11 minutes (range, 0–2 hours 30 minutes). The mean total time (including traveling and waiting time and time spent with the clinician) for an ART visit was 3 hours 17 minutes (range, 0–21 hours 2 minutes).

Regarding the indirect costs, mean potential patient costs related to total time lost during travel, waiting and clinician contact multiplied by the estimated participants' income for ART and PMTCT patients were US$3.43 and US$3.53 per visit (see Table 5). Based on a minimum sectoral wage of US$74 and the high skilled average wage of US$3.14 per hour; the mean indirect cost for ART and PMTCT ranged from US$1.11 to US$8.53 for ART patients and US$1.13 to US$8.69 for PMTCT patients respectively.

## Discussion

The overall mean annual patient costs per person for PMTCT (US$20.00) and ART (US$19.31) in this evaluation. While these costs are at least 2% of the minimum sectoral wage (US$75), they are 79% and 76% of Zimbabwe's own allocated health expenditure per capita budget of US$25.45 [11], reflecting that the cost of PMTCT and ART services in Zimbabwe can barely be covered under the budget. However, per unit costs for certain services showed high values when restricted to those who accessed a certain service. Without a general improvement of the socio-economic conditions in communities with the highest risk of HIV, HIV elimination will be challenging [12]. Financial barriers hinder access to health services in low- and middle-income countries [12, 13] and are usually related to out-of-pocket expenses and the impact on household budgets [14]. The World Health Organization [15] states that when people suffer financial hardship due to out-of-pocket expenses, universal health coverage cannot be achieved.

Although ART and PMTCT services and their associated diagnostics are offered free of charge within the public sector in Zimbabwe, access to such services is still a challenge in a country that has experienced myriad of economic hardships over the past decade. We found that patients are still incurring costs. Among PMTCT patients who had costs associated inpatient admissions, the costs were large (mean = $216.36, range = $24.00–$1200.00, n = 11). Although PMTCT visits are monthly at the beginning of the pregnancy, the number of visits tends to decrease during the postnatal period, contributing to reduced mean transport costs.

Although the potential patient costs related to time lost per visit ranged between US$1.10 to more than US$8 for either PMTCT and ART patients, this adds on average, a 1% burden of a monthly wage [10]. Although seemingly a small amount, prolonged absenteeism from one's economic activities while seeking HIV-related care can result in loss of earnings which adds to the financial woes to an already financially overburdened populace.

This evaluation had several limitations. The analysis did not include all indirect costs (i.e., productivity losses) associated with HIV-related morbidity and mortality. This evaluation also did not include information from patients regarding the services they did not receive due to financial constraints. This evaluation also primarily focused on costs incurred by patients who received services from publicly funded facilities and therefore did not estimate costs for patients solely receiving services at private, for-profit clinics. In addition, the patient costs for receiving PMTCT and ART services were self-reported and may not be entirely accurate. As a result, there is likely to be recall bias. The study sample may also not be representative of all PMTCT and ART patients in Zimbabwe.

Future studies should evaluate the impacts of incurred costs on the ability of patients to access care and how DSD models could help address these issues. As new policies and guidelines are implemented, public health leaders should consider patient expenses as a potential barrier to accessing care.

## Supporting information

**S1 Data.**
(DTA)

## Acknowledgments

We would like to thank the numerous people and organizations involved in the main evaluation, from which the manuscript was developed, without whom we would not have been able to successfully complete it. We would like to acknowledge the generous contributions of time and effort to the staff members of Ministry of Health and Child Care and their key implementing partners, the U.S Centers for Disease Control and the Ministry of Public Works, Local Government and National Housing, the PACE Technical Working Group and the data collection team. We would also like to recognize and acknowledge the efforts of all the authors and reviewers of this manuscript for enhancing the quality of the final product.

We would also like to recognize and acknowledge the efforts of the staff members of the 20 facilities involved in the evaluation: Parirenyatwa, United Bulawayo Hospital, Mutare Provincial Hospital, Masvingo Provincial Hospital, Nyanga District Hospital, Filabusi District Hospital, Mount Selinda, Gutu Mission Hospital, Kadoma General Hospital, Kwekwe General Hospital, Weya Rural Hospital, Lundi Rural Hospital, St Albert's Mission hospital, Mashoko Mission Hospital, Garaba Rural Health Centre, Dendera Rural Health Centre, Hatfield Clinic, Rujeko Clinic, Chipararwe Council Clinic, Nkankezi Rural Health Centre, National Microbiology Reference Laboratory and Mpilo Hospitals. Last but not least, we gratefully acknowledge the men and women who responded to our patient questionnaire. Special mention also goes to Ms. Leala Ruangtragool, PHI/CDC Global Health fellow, for her technical support throughout the PACE implementation process.

**Disclaimer:** The findings and conclusions in this publication are those of the authors and do not necessarily represent the official position of the funding agencies or any organization represented.

## Author Contributions

**Conceptualization:** Innocent Chingombe, Munyaradzi P. Mapingure, Shirish Balachandra, Tendayi N. Chipango, Angela Mushavi, Tsitsi Apollo, Chutima Suraratdecha, John H. Rogers, Leala Ruangtragool, Elizabeth Gonese, Godfrey N. Musuka, Owen M. Mugurungi, Tiffany G. Harris.

**Formal analysis:** Munyaradzi P. Mapingure, Tendayi N. Chipango.

**Funding acquisition:** Innocent Chingombe, Shirish Balachandra, John H. Rogers, Leala Ruangtragool, Elizabeth Gonese, Godfrey N. Musuka, Tiffany G. Harris.

**Investigation:** Munyaradzi P. Mapingure, Shirish Balachandra, Angela Mushavi, Tsitsi Apollo, Chutima Suraratdecha, John H. Rogers, Leala Ruangtragool, Elizabeth Gonese, Godfrey N. Musuka, Owen M. Mugurungi, Tiffany G. Harris.

**Methodology:** Innocent Chingombe, Munyaradzi P. Mapingure, Shirish Balachandra, Tendayi N. Chipango, Tsitsi Apollo, Chutima Suraratdecha, John H. Rogers, Leala Ruangtragool, Elizabeth Gonese, Godfrey N. Musuka, Owen M. Mugurungi, Tiffany G. Harris.

**Project administration:** Innocent Chingombe, Munyaradzi P. Mapingure, Shirish Balachandra, Angela Mushavi, Tsitsi Apollo, Chutima Suraratdecha, John H. Rogers, Leala Ruangtragool, Elizabeth Gonese, Godfrey N. Musuka, Owen M. Mugurungi.

**Resources:** Shirish Balachandra, Tsitsi Apollo, Chutima Suraratdecha, John H. Rogers, Elizabeth Gonese, Godfrey N. Musuka, Tiffany G. Harris.

**Software:** Innocent Chingombe.

**Supervision:** Innocent Chingombe, Shirish Balachandra, Tendayi N. Chipango, Angela Mushavi, Tsitsi Apollo, Chutima Suraratdecha, John H. Rogers, Leala Ruangtragool, Elizabeth Gonese, Godfrey N. Musuka, Owen M. Mugurungi.

**Validation:** Innocent Chingombe, Shirish Balachandra, Fiona Gambanga, John H. Rogers, Elizabeth Gonese, Godfrey N. Musuka, Owen M. Mugurungi, Tiffany G. Harris.

**Writing – original draft:** Innocent Chingombe, Tendayi N. Chipango, Fiona Gambanga, Chutima Suraratdecha, John H. Rogers, Leala Ruangtragool, Elizabeth Gonese, Godfrey N. Musuka, Tiffany G. Harris.

**Writing – review & editing:** Innocent Chingombe, Tendayi N. Chipango, Fiona Gambanga, Angela Mushavi, Tsitsi Apollo, Chutima Suraratdecha, John H. Rogers, Leala Ruangtragool, Elizabeth Gonese, Godfrey N. Musuka, Owen M. Mugurungi, Tiffany G. Harris.

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
