## [Decision Letter · Decision Letter 0]

23 Mar 2021

PONE-D-21-04638

Patient costs for prevention of mother-to-child HIV transmission and antiretroviral therapy services in public health facilities in Zimbabwe

PLOS ONE

Dear Dr. Chingombe,

Thank you for submitting your manuscript to PLOS ONE. After careful consideration, we feel that it has merit but does not fully meet PLOS ONE’s publication criteria as it currently stands. Therefore, we invite you to submit a revised version of the manuscript that addresses the points raised during the review process.

Please review comments below and address. Please note PLOS ONE data policy. For the third party data not owned by the authors, PLOS ONE requires that the authors provide all information necessary (e.g. URLs, contact information, dataset titles, etc) for other researchers to access the same data used to present the findings of their study and to confirm that they accessed the data in the same manner they expect future researchers to do so, and did not receive special privileges from the third party.  It is not sufficient to note that data is available with the Zimbabwe Ministry of Health. Further, if the data are owned by the authors, PLOS ONE requires the data to be made available unless there are specific legal or ethical considerations for doing so. 

We look forward to receiving your revised manuscript.

Kind regards,

Saeed Ahmed

Academic Editor

PLOS ONE

Journal Requirements:

2. Please provide additional details regarding participant consent. In the ethics statement in the Methods and online submission information, please ensure that you have specified how verbal consent was documented and witnessed.

3. In your Methods section, please provide additional information about the participant recruitment method and the demographic details of your participants. Please ensure you have provided sufficient details to replicate the analyses such as:

a) the recruitment date range (month and year),

b) a description of any inclusion/exclusion criteria that were applied to participant recruitment,

c) a table of relevant demographic details,

d) a statement as to whether your sample can be considered representative of a larger population,

e) a description of how participants were recruited, and

f) descriptions of where participants were recruited and where the research took place.

4. Please include additional information regarding the survey or questionnaire used in the study and ensure that you have provided sufficient details that others could replicate the analyses.

For instance, if you developed a questionnaire as part of this study and it is not under a copyright more restrictive than CC-BY, please include a copy, in both the original language and English, as Supporting Information. Moreover, please include more details on how the questionnaire was pre-tested, and whether it was validated.

**Comments to the Author**

1. Is the manuscript technically sound, and do the data support the conclusions?

Reviewer #1: Yes

Reviewer #2: Yes

2. Has the statistical analysis been performed appropriately and rigorously? 

Reviewer #1: Yes

Reviewer #2: Yes

3. Have the authors made all data underlying the findings in their manuscript fully available?

Reviewer #1: Yes

Reviewer #2: No

4. Is the manuscript presented in an intelligible fashion and written in standard English?

Reviewer #1: Yes

Reviewer #2: Yes

5. Review Comments to the Author

Reviewer #1: The authers have used standard costing methodology, though need more explanations, to estimate costs associated with seeking and receiving PMCT and ART care in Zimbabwe. Costs were summarized using means and medians and ranges appropriately described.

English is good and clear and manuscript is well presented

Reviewer #2: This is a nice study with contact with a considerable sample of ART and PMTCT cohort in Zimbabwe. It demonstrates a cost to individuals to access "free" health care services.

There are a few areas where perhaps more clarification can be made to outline procedures for those unfamiliar with clinical processes in Zimbabwe. The site selection process that removed poorly functioning clinics poses some potential bias as clients at those sites likely occur much more cost if the clinic is not run well, and its unclear the size of those poorly functioning clinics to know if they represent a large part of the national cohort, as it represents some percentage of the 565 excluded clinics - 1/3 of all sites in the country.

If included in the questions to clients, often in addition to wages lost many clients lose employment opportunities due to the recurrent absences they need to incur to attend ART appointments. Were clients asked if they have lost jobs due to recurrent absences?

Additionally the authors discuss client visits with "appointments" versus "without appointment" visits and its unclear if in Zimbabwe these "appointments" are for a certain day or if they are given a time slot within the day which may affect the amount of time spent at the facility. For example, are clients given an appointment on 5 April or 5 April at 1400.

Further details of the selection process for phone interview will be helpful as clinic staff identified clients eligible for phone interview had to include only those who had phones and it's not clear what percentage of the ART/PMTCT cohort in Zimbabwe has a phone available for followup. It seems unusual that the authors could not find enough clients who missed even one appointment so perhaps they couldn't find enough who also had a phone?

The authors report, "The data underlying the results presented in the study are available from Zimbabwe

Ministry of Health and Child Care." I am not sure if that qualifies as readily available and defer to the editors.

6. PLOS authors have the option to publish the peer review history of their article (what does this mean?). If published, this will include your full peer review and any attached files.

Reviewer #1: No

Reviewer #2: No

---

## [Author Response · Author response to Decision Letter 0]

11 May 2021

ICAP at Columbia University, Zimbabwe Country Office

107 King George 

Avondale

Harare

Zimbabwe

Editorial Office

PLOS ONE

Re: Edits/Response to Reviewers for “Patient costs for prevention of mother-to-child HIV transmission and antiretroviral therapy services in public health facilities in Zimbabwe”

Dear Editor

We appreciate the opportunity to resubmit our work to your journal. We would like to thank you for helpful comments and suggestions. The manuscript has been modified accordingly, with, revisions in tracked changes. Responses to the reviewers’ suggestions are given below, in bold font.

Editor’s Comment

Please review comments below and address. Please note PLOS ONE data policy. For the third party data not owned by the authors, PLOS ONE requires that the authors provide all information necessary (e.g. URLs, contact information, dataset titles, etc) for other researchers to access the same data used to present the findings of their study and to confirm that they accessed the data in the same manner they expect future researchers to do so, and did not receive special privileges from the third party. It is not sufficient to note that data is available with the Zimbabwe Ministry of Health. Further, if the data are owned by the authors, PLOS ONE requires the data to be made available unless there are specific legal or ethical considerations for doing so. 

Response: We have provided the details in the online submission manager of how data can be accessed. i.e. “Data are available from the MoHCC, as STATA Format named PACE Study Patient Data. Contact Dr Brian Moyo , Epidemiologist at the Zimbabwe Ministry of Health’s AIDS and TB Unit, e-mail- moyobk1@gmail.com. Phone +263 774021723..”

Comment 1. Please ensure that your manuscript meets PLOS ONE's style requirements, including those for file naming. The PLOS ONE style templates can be found at

Response: The manuscript has been revised to match the PLOS ONE’s style requirements. 

Comment 2. Please provide additional details regarding participant consent. In the ethics statement in the Methods and online submission information, please ensure that you have specified how verbal consent was documented and witnessed.

We have elaborated in the manuscript on page 6-7 on the details of participant consent. Briefly, to obtain informed oral consent, interviewers read the consent form to eligible potential participants in the language that was preferable to the individual and provided them the opportunity to ask any questions. The consent form covered all procedures, potential risks and benefits, and who to contact to report complaints or concerns. After potential participants indicated that they clearly understood the content of the informed consent form, they provided verbal consent by repeating the consenting statement. Those not willing to take part, would read a refusal statement. The interviewer signed an attestation that consent was provided. Those that were enrolled in person were provided a copy of the consent form. 

Comment 3. In your Methods section, please provide additional information about the participant recruitment method and the demographic details of your participants. Please ensure you have provided sufficient details to replicate the analyses such as:

a) the recruitment date range (month and year),

The recruitment date ranged from September-November 2017 as written in the first sentence on page 6.

b) a description of any inclusion/exclusion criteria that were applied to participant recruitment,

The inclusion and exclusion criteria is now reflected on page 6. Briefly, at each site, we selected a cross-sectional convenience sample among adult patients (aged ≥18 years) receiving either outpatient PMTCT services in ANC (pregnant women and postpartum women with HIV) or ART services (ART and pre-ART patients) at the facility on the day of data collection and for whom the date of encounter was not the first visit at the facility. Exclusion criteria used included individuals who were not enrolled in the PMTCT or ART programs at the site, individuals aged less than 18 years of age, individuals who were newly registered PMTCT or ART patients in the health facility to ensure that individuals in the sample have had experience with PMTCT or ART services in the facility, individuals who do not consent to be interviewed in English, Shona, or Ndebele.

c) a table of relevant demographic details,

Table 1 on page 9 already includes the demographics details.

d) a statement as to whether your sample can be considered representative of a larger population,

We have added in the limitations section on page 16 that “This study sample may not be representative of all PMTCT and ART patients in Zimbabwe”.

e) a description of how participants were recruited, and

This is already described in the Facilities and Participants subsections on page 5 and 6.

f) descriptions of where participants were recruited and where the research took place.

This is already described in the Facilities and Participants subsections on page 5 and 6.

Comment 4. Please include additional information regarding the survey or questionnaire used in the study and ensure that you have provided sufficient details that others could replicate the analyses. 

For instance, if you developed a questionnaire as part of this study and it is not under a copyright more restrictive than CC-BY, please include a copy, in both the original language and English, as Supporting Information. Moreover, please include more details on how the questionnaire was pre-tested, and whether it was validated.

The evaluation made use of a standard patient questionnaire developed by CDC, which they used in other countries where they have supported similar studies. It was adapted to the local context and translated to Shona and Ndebele languages. 

Comment 5. We note that you have indicated that data from this study are available upon request. PLOS only allows data to be available upon request if there are legal or ethical restrictions on sharing data publicly. For information on unacceptable data access restrictions, please see http://journals.plos.org/plosone/s/data-availability#loc-unacceptable-data-access-restrictions.

Data can be accessed in writing, from the MOHCC Epidemiologist Dr Brian Moyo, who is based at MOHCC AIDS and TB Programme office at 2nd floor Mkwati Building in Harare, via the following e-mail, moyobk1@gmail.com. Phone +263 774021723. Data are available from the MoHCC, as STATA Format named PACE Study Patient Data

Comment 6. In your revised cover letter, please address the following prompts:

 Upon release of the final report, third-party researchers may e-mail one of the MOHCC PIs listed in the protocol to gain access to the processed data. Permission given to any third party shall be communicated in writing, stating the terms and conditions that need to be observed by the party when using the data, they would have been given access to.

Data can be accessed in writing, from the MOHCC Epidemiologist Dr Brian Moyo, who is based at MOHCC AIDS and TB Programme office at 2nd floor Mkwati Building in Harare, via the following e-mail, moyobk1@gmail.com. Phone +263 774021723. 

Comment 7. b) If there are no restrictions, please upload the minimal anonymized data set necessary to replicate your study findings as either Supporting Information files or to a stable, public repository and provide us with the relevant URLs, DOIs, or accession numbers. Please see http://www.bmj.com/content/340/bmj.c181.long for guidelines on how to de-identify and prepare clinical data for publication. For a list of acceptable repositories, please see http://journals.plos.org/plosone/s/data-availability#loc-recommended-repositories.

Ministry of Health and Child Care will provide data upon request and as mentioned above, we have provided the details in the online submission manager of how data can be accessed. i.e. “Data are available from the MoHCC, as STATA Format named PACE Study Patient Data. Contact Dr Brian Moyo, moyobk1@gmail.com. Phone +263 774021723.

Comment 8. PLOS requires an ORCID iD for the corresponding author in Editorial Manager on papers submitted after December 6th, 2016. Please ensure that you have an ORCID iD and that it is validated in Editorial Manager. To do this, go to ‘Update my Information’ (in the upper left-hand corner of the main menu), and click on the Fetch/Validate link next to the ORCID field. This will take you to the ORCID site and allow you to create a new iD or authenticate a pre-existing iD in Editorial Manager. Please see the following video for instructions on linking an ORCID iD to your Editorial Manager account: https://www.youtube.com/watch?v=_xcclfuvtxQ

The ORCID iD for the first author is https://orcid.org/0000-0001-6512-2832. The paper will be linked to the first author ORCID iD when re-submitting the revised version.

Comment 9. Please review your reference list to ensure that it is complete and correct. If you have cited papers that have been retracted, please include the rationale for doing so in the manuscript text, or remove these references and replace them with relevant current references. Any changes to the reference list should be mentioned in the rebuttal letter that accompanies your revised manuscript. If you need to cite a retracted article, indicate the article’s retracted status in the References list and also include a citation and full reference for the retraction notice.

Our reference list have been revised to match the expected standards.

All authors concur with the re-submission. This work has neither been published nor is it simultaneously being considered for publication elsewhere. Neither I, nor any of my co-authors, have any conflicting interests. 

We look forward to hearing from you regarding whether or not these revisions have improved the manuscript such that the revised version can be published in PLOS ONE.

Yours sincerely, 

Innocent Chingombe

---

## [Editor Report · Decision Letter 1]

10 Jun 2021

PONE-D-21-04638R1

Patient costs for prevention of mother-to-child HIV transmission and antiretroviral therapy services in public health facilities in Zimbabwe

PLOS ONE

Dear Dr. Chingombe,

Thank you for submitting your manuscript to PLOS ONE. After careful consideration, we feel that it has merit but does not fully meet PLOS ONE’s publication criteria as it currently stands. Therefore, we invite you to submit a revised version of the manuscript that addresses the points raised during the review process.

ACADEMIC EDITOR:

We have been trying to contact you to address comments made by reviewers. Please make sure to respond to all comments in a point-by-point manner in your response letter.  Comments from reviewer 1 are attached and comments from Reviewer 1 and 2 are below.

We look forward to receiving your revised manuscript.

Kind regards,

Saeed Ahmed

Academic Editor

PLOS ONE

Journal Requirements:

**Review comments from first round of reviews:**

**Reviewer #1**: The authers have used standard costing methodology, though need more explanations, to estimate costs associated with seeking and receiving PMCT and ART care in Zimbabwe. Costs were summarized using means and medians and ranges appropriately described.

English is good and clear and manuscript is well presented

**Reviewer #2:** This is a nice study with contact with a considerable sample of ART and PMTCT cohort in Zimbabwe. It demonstrates a cost to individuals to access "free" health care services.

There are a few areas where perhaps more clarification can be made to outline procedures for those unfamiliar with clinical processes in Zimbabwe. The site selection process that removed poorly functioning clinics poses some potential bias as clients at those sites likely occur much more cost if the clinic is not run well, and its unclear the size of those poorly functioning clinics to know if they represent a large part of the national cohort, as it represents some percentage of the 565 excluded clinics - 1/3 of all sites in the country.

If included in the questions to clients, often in addition to wages lost many clients lose employment opportunities due to the recurrent absences they need to incur to attend ART appointments. Were clients asked if they have lost jobs due to recurrent absences?

Additionally the authors discuss client visits with "appointments" versus "without appointment" visits and its unclear if in Zimbabwe these "appointments" are for a certain day or if they are given a time slot within the day which may affect the amount of time spent at the facility. For example, are clients given an appointment on 5 April or 5 April at 1400.

Further details of the selection process for phone interview will be helpful as clinic staff identified clients eligible for phone interview had to include only those who had phones and it's not clear what percentage of the ART/PMTCT cohort in Zimbabwe has a phone available for followup. It seems unusual that the authors could not find enough clients who missed even one appointment so perhaps they couldn't find enough who also had a phone?

The authors report, "The data underlying the results presented in the study are available from Zimbabwe

Ministry of Health and Child Care." I am not sure if that qualifies as readily available and defer to the editors.
---

## [Author Response · Author response to Decision Letter 1]

1 Jul 2021

Editorial Office

PLOS ONE

Re: Edits/Response to Reviewers for “Patient costs for prevention of mother-to-child HIV transmission and antiretroviral therapy services in public health facilities in Zimbabwe”

Dear Editor

We appreciate the opportunity to resubmit our work to your journal. We would like to thank you for helpful comments and suggestions. The manuscript has been modified accordingly, with, revisions in tracked changes. We had missed some comments and have now responded to the reviewers’ suggestions as detailed below, in bold italicised font.

Reviewer #1:

The authors objectives were to estimate costs associated with seeking and receiving PMCT and ART treatment from a household perspective. This is an interesting paper for programmers and policy makers providing care for people living with HIV and AIDs.

Thank you

Comment 1: The authors should be explicit about the costing approach used and describe it in more detail. It appears a bottom up or micro costing methodology was used but there is no reference to this

We have added the section with a description on the methodology on page 8 of the revised manuscript

Comment 2: What are the implication of using potential earnings to input for lost time? If unemployment is high in Zimbabwe implying not much opportunity costs for the time loss, this would bias the cost estimates upwards. The Authors must critically comment about this. A sensitivity analysis would have been in order to explore uncertainty surrounding such assumptions.

We aligned the income based on the information on income sources provided by the participants. We drew the income information from several sources. As part of the sensitivity analysis, we measured minimal wage and maximum casual wage rate. As the economic crises progresses in Zimbabwe; the health system has to advance to accommodate the need to earn money by patients.

Comment 3: Please present summary table of the costs, by the standard cost categories i.e. medical, non-medical and indirect costs etc(Refer to Drummond). Present these separately for PMCT and ART. By comparing the two, you could gain some insight if PMCT cost profile is markedly different from ART profile, which may be useful. For example, PMCT women are generally not very sick and may walk to and from ANC compared to ART patients who are often sick, have others accompanying them and my use different and more costly modes of transport. And in view of this, can author explain why the mean time to care for PMCT and ART are not substantially different, I would expect this given the explanation above.

We have presented these in the table. The cost difference between the groups was very small. With an ART profile-people are being put into treatment earlier especially since we didn’t look at first visits. We also focussed on refill visits to the specific service centres versus outpatient visits in an emergency situation. We did account for such visits but they were not the primary focus and were not reported as frequently.

Comment 4: The costs are described as low (page 13) begging the question compared to what?

We have compared to the household monthly income and health sector budget per capita.

Comment 5: It would be useful to express the annual costs a share of total annual household income or expenditure on health, this would give an idea about burden which can be compared with other available data published in the literature.

Comment 6: Limitation section should include re-call biased, as patients are likely to accurately report costs for recent than relatively more distant events or illness episodes. One can do quick robust checks to test this premise and report on findings

This has been added 

Comment 7: Selection bias is another limitation, as only individuals seeking care in facilities were interviewed, so results can not be generalised to the general populations. The author can use Heckman models to estimate their costs and control for this bias.

We tried to obtain information from those who did not attend visits but the sample size was not large enough

Reviewer #2:

This is a nice study with contact with a considerable sample of ART and PMTCT cohort in Zimbabwe. It demonstrates a cost to individuals to access "free" health care services. 

Thank you

Comment 1: There are a few areas where perhaps more clarification can be made to outline procedures for those unfamiliar with clinical processes in Zimbabwe. The site selection process that removed poorly functioning clinics poses some potential bias as clients at those sites likely occur much more cost if the clinic is not run well, and its unclear the size of those poorly functioning clinics to know if they represent a large part of the national cohort, as it represents some percentage of the 565 excluded clinics - 1/3 of all sites in the country.

More clarifications have been added to the methodology

Comment 2: If included in the questions to clients, often in addition to wages lost many clients lose employment opportunities due to the recurrent absences they need to incur to attend ART appointments. Were clients asked if they have lost jobs due to recurrent absences? 

We have included further clarifications on these calculations, see page 8

Comment 3: Additionally the authors discuss client visits with "appointments" versus "without appointment" visits and its unclear if in Zimbabwe these "appointments" are for a certain day or if they are given a time slot within the day which may affect the amount of time spent at the facility. For example, are clients given an appointment on 5 April or 5 April at 1400. 

Appointments are per day and do not drill down to the time

Comment 4: Further details of the selection process for phone interview will be helpful as clinic staff identified clients eligible for phone interview had to include only those who had phones and it's not clear what percentage of the ART/PMTCT cohort in Zimbabwe has a phone available for followup. It seems unusual that the authors could not find enough clients who missed even one appointment so perhaps they couldn't find enough who also had a phone?

This is described on page 7 of the manuscript. Because participants enrolled on-site may visit the facility more frequently, resulting in overestimation of patient costs (since more frequent facility visits entail higher costs), the evaluation also included phone interviews for adult ART or PMTCT patients who had at least one prior appointment at the facility and had missed their most recent appointment

Comment 5: The authors report, "The data underlying the results presented in the study are available from Zimbabwe Ministry of Health and Child Care." I am not sure if that qualifies as readily available and defer to the editors.

We uploaded all the required data

All authors concur with the re-submission. This work has neither been published nor is it simultaneously being considered for publication elsewhere. Neither I, nor any of my co-authors, have any conflicting interests. 

We look forward to hearing from you regarding whether or not these revisions have improved the manuscript such that the revised version can be published in PLOS ONE.

Yours sincerely, 

Innocent Chingombe

---

## [Decision Letter · Decision Letter 2]

4 Aug 2021

Patient costs for prevention of mother-to-child HIV transmission and antiretroviral therapy services in public health facilities in Zimbabwe

PONE-D-21-04638R2

Dear Dr. Chingombe,

We’re pleased to inform you that your manuscript has been judged scientifically suitable for publication and will be formally accepted for publication once it meets all outstanding technical requirements.

Kind regards,

Saeed Ahmed

Academic Editor

PLOS ONE

---

## [Editor Report · Acceptance letter]

9 Aug 2021

PONE-D-21-04638R2 

Patient costs for prevention of mother-to-child HIV transmission and antiretroviral therapy services in public health facilities in Zimbabwe 

Dear Dr. Chingombe:

I'm pleased to inform you that your manuscript has been deemed suitable for publication in PLOS ONE. Congratulations! Your manuscript is now with our production department. 

Kind regards, 

on behalf of

Dr. Saeed Ahmed 

Academic Editor

PLOS ONE